# Perfectionism and Pain Intensity in Women with Fibromyalgia: Its Influence on Activity Avoidance from The Contextual Perspective

**DOI:** 10.3390/ijerph17228442

**Published:** 2020-11-14

**Authors:** Blanco Sheila, Luque-Reca Octavio, Catala Patricia, Bedmar Dolores, Velasco Lilian, Peñacoba Cecilia

**Affiliations:** 1Department of Psychology, Universidad Rey Juan Carlos, 28933 Madrid, Spain; sheila.blanco.rico@urjc.es (B.S.); octavio.luque@urjc.es (L.-R.O.); patricia.catala@urjc.es (C.P.); lilian.velasco@urjc.es (V.L.); 2Pain Unit, Hospital Universitario de Fuenlabrada, 28944 Fuenlabrada, Madrid, Spain; mariadolores.bedmar@salud.madrid.org

**Keywords:** fibromyalgia, perfectionism, pain, activity avoidance

## Abstract

Given the scarcity of studies regarding perfectionism from a contextual perspective, this study aims to analyze its role in the relationship between pain and activity avoidance and its differential effect among patients with different fibromyalgia severity. A cross-sectional study with 228 women with fibromyalgia classified into two disease severity groups (low/moderate vs. high) was carried out. Moderation analyses were conducted; perfectionism was used as moderator, pain (in high and low pain situations) as independent variable, and activity avoidance as the outcome. Among the high disease severity group, analyses showed direct contributions of perfectionism (*p* < 0.001) but not of pain (*p* > 0.05); moderation effects were found in high pain situations (*p* = 0.002) (for low levels of perfectionism, a positive association was found between pain intensity and avoidance). Among the low severity group, direct effects of perfectionism (*p* < 0.05) and pain intensity (*p* = 0.04) were found (although the latter only for high pain situations); moderation effects were found in high pain situations (*p* = 0.018) (for high levels of perfectionism a positive and significant association was found between pain intensity and avoidance). Perfectionism has been found to be a key variable in the differential relationship between pain intensity (in high pain situations) and activity avoidance in groups with high and low disease severity.

## 1. Introduction

Fibromyalgia (FM) is a complex syndrome characterized by frequent chronic musculoskeletal pain and widespread sensitivity, which is often accompanied by symptoms such as extreme fatigue [1], cognitive decline [2], sleep disturbances [3], and psychological and affective disorders [4], among others [5,6]. These symptoms cause significant functional impairment [7] and higher levels of physical deterioration [8]. Widespread pain is the cardinal symptom in fibromyalgia and has a major effect on quality of life. Furthermore, pain (diagnosed either generically or through tender points) has been found to have a close relationship with the impact of the disease [9] and has been used in different studies as an indicator of disease severity [10]. Pain as the main symptom in fibromyalgia is clearly mentioned in the initial diagnostic criteria [5], although these have evolved into the current standards of diagnosis [6,11], and now other additional symptoms are also contemplated (e.g., fatigue, non-restorative sleep, cognitive disorders). Both diagnostic criteria continue to coexist today, and previous literature has shown a good agreement between them for FM diagnosis [5,6,12].

### 1.1. Activity Avoidance, Fear of Movement and Disability

The complex symptoms of fibromyalgia, including pain (disease severity), interfere in patients’ daily function, preventing them from accomplishing their goals and reducing their ability to plan work and social activities [13]. Because of this, one of the main treatment aims in patients with fibromyalgia is to include or maintain physical function and to avoid further disability [14,15]. Fibromyalgia treatments that have included activities and physical exercise as one of their therapeutic aims have shown positive effects on patients’ health outcomes [16,17,18]. However, people with fibromyalgia often show lower levels of physical activity and greater sedentary behaviors [19,20,21] due to pain and fatigue that are often initially exacerbated with increased activity [22]. Traditionally, activity avoidance has been explained within the framework of fear of movement/(re) injury [23] (see Figure 1). This model explains why the disability may become chronic, as it explains the maintenance or the exacerbation of the fear. Activity avoidance could possibly result in a phobic state, with the patient no longer being able to perform certain activities because they anticipate that certain activities may increase pain and suffering. This could lead to self-perpetuating cycles based on prolonged avoidance of motor activities and pain avoidance goals. This behavioral pattern is a maladaptive way of coping with pain [24] with harmful consequences, both physically and psychologically [25].

### 1.2. Perfectionism and Fibromyalgia. A Contextual-Functional Approach

More recently, from a contextual functional approach, the model of psychological flexibility [26] has been proposed as a unifying framework for understanding activity patterns, including avoidance and persistence. This model postulates that there is no linearity between the two variables and even that the modulators themselves can depend on contextual variables that influence committed actions towards goals. Therefore, knowing the context in which the behavior occurs is important to determine its influence on functioning [26]. In this sense, previous studies have suggested that pain is a key contextual variable to be taken into account [27]. Furthermore, extensive research has pointed to the important role of psychological factors when explaining physical disability and mental well-being, with pain catastrophizing as one of the most studied factors [27]. In this context, perfectionism has been analyzed as a very stressful cognitive-personality vulnerability factor [28,29] frequent in patients with fibromyalgia [4,30,31].

Based on this model, perfectionism is defined by cognitive rigidity and behavioral inflexibility [32,33], where the person presents lack of situation strategy and uses the same strategies [34] regardless of the context. One of the underlying mechanisms in perfectionism is experiential avoidance [35], which influences health outcomes. For this reason, acceptance and commitment therapy has been used [33,35] to reduce perfectionism and the associated symptoms. Previous studies have shown that there is an association between perfectionism and stress, poor mental health, reductions in functioning, and the frequency/intensity of pain and fatigue [36,37,38]. Particularly, perfectionism is associated with reductions in health functioning in women with fibromyalgia [37], mediated by behavioral disengagement processes, denial, and self-blame [29,30].

The scarce previous literature in this area has shown that, in young patients with chronic pain, perfectionism can have indirect effects on functional limitation and other negative consequences of pain, depending on biopsychosocial variables (especially pain catastrophizing and pain-related fear) [39,40]. However, to date, no studies have analyzed the role of perfectionism in the relationship between pain and avoidance in women with fibromyalgia. Because of this, the main objective of this study was to analyze possible moderating mechanisms. The present study aimed to analyze the role of perfectionism as a moderator in the relationship between pain intensity (in situations with high or low pain intensity) and activity avoidance and their differential effect on patients with high or low/moderate disease severity in women with fibromyalgia. Thus, consistent with past research on the moderating role of pain [27], we anticipated that perfectionism would be a moderating factor in the relationship between pain intensity and activity avoidance in fibromyalgia patients. Furthermore, it was hypothesized that this role would vary depending on the degree of the disease severity and the severity of pain in the specific situation. Particularly, we expected a direct effect of perfectionism on activity avoidance, regardless of the severity of pain and disease, since perfectionism creates rigid avoidance behaviors [33,35]. Finally, we expected that the moderating effect of perfectionism on the pain intensity would have a greater influence on avoidance behaviors in the high disease severity group because these patients maintain a high level of pain over time, which favors a greater goal conflict between the avoidance of pain [7,8] and the fulfillment of tasks relevant to them [32,35]. The findings might have important repercussions on the design of programs to improve adherence to physical exercise, attending to the heterogeneity of these patients [38].

## 2. Materials and Methods

### 2.1. Participants

In the study, 228 Spanish women with fibromyalgia were recruited from different Fibromyalgia Associations according to the American College of Rheumatology (ACR) criteria [5,6]. The study followed the ethical principles for research with human participants and was approved by the University Committee on Ethics (Registration number: 160520165916; PI17/00858). Participants completed a booklet of questionnaires that took 15–20 min after having signed the informed consent to participate in the project. Patients did not receive any economic compensation for participating in the study.

Due to the heterogeneity and the differential profiles of fibromyalgia patients [38,41], the sample was divided into two groups according to pain severity as an indicator of disease severity. As has been pointed out, the use of pain as an indicator of the severity of the disease is supported by its diagnostic role in the criteria most used today [5] and its close correlation with the most current criteria [12]. Although other indicators of disease severity could be used, pain severity has shown its close relationship with other types of associated symptoms in this population, including emotional symptoms and functional limitation [9], and has been used as an indicator of severity in previous studies [10]. However, given the diversity of existing indicators of severity disease in fibromyalgia patients, the results must be interpreted while taking this decision into account. Following the methodology used in previous studies [8,42], a score of 7 or higher in pain intensity was considered as indicative of belonging to the high severity group, whilst scores below 7 allocated patients in the low/moderate severity group. Patients were classified into these two groups based on their scores on the pain item of the Revised Fibromyalgia Impact Questionnaire (FIQR; item 12: “please indicate the intensity of the pain in the last 7 days, where 0 is no pain and 10 is unbearable pain”). The distribution of the participants into the two groups was as follows: high severity group (*n* = 124) and low/moderate severity group (*n* = 104). To determine the sample size required in these two groups for regression and moderation analyses proposed, the G*Power software [43] was used to perform a-priori power analysis. Taking three predictor variables, a power of 80%, an intermediate effect size, and an alpha level of 0.05, we obtained that a minimum of 77 participants would be necessary in each group.

No statistically significant differences were found between high (*n* = 124) and low/moderate severity (*n* = 104) groups for sociodemographic variables, except for work status (*p* = 0.038). Regarding this variable, 8.62% of the high severity group were employed compared to 20% of the low/moderate severity group. The percentages of sick leaves and patients who were unemployed were lower among the low/moderate severity group than among the high severity group (3.33% vs. 12.64% and 6.66% vs. 14.37% respectively). The percentages of retired and housewives were similar among both groups (approximately 35%).

### 2.2. Measures

Activity avoidance: The Spanish version of Activity Patterns Scale [44] was used to assess activity avoidance. This is a questionnaire with 24 items ranged on a 5-point Likert response scale with anchors 0 = “never” to 4 = “always”. The activity avoidance dimension, which is composed of 3 items, was selected for the present study purposes. The other seven dimensions in the questionnaire are pain avoidance, task-contingent persistence, excessive persistence, pain-contingent persistence, pacing to increase activity levels, pacing to conserve energy for valued activities, and pacing to reduce pain. Higher scores indicate greater presence of activity avoidance. The internal consistency of the activity avoidance dimension in the present study was 0.70.

Perfectionism: The Spanish version of the Frost Multidimensional Perfectionism Scale (FMPS) [45] was used. The FMPS consists of 35 items with a 5-point Likert (1–5) scale grouped into 6 factors (Personal Standards, Concern over Mistakes, Doubt about Actions, Parental Expectations, Parental Criticism, and Organization) and total score. Perfectionism is the belief that perfection can and should be achieved. Perfectionists value themselves based on their self-imposed achievements. Therefore, the “Concern over Mistakes” subscale (reflecting negative reactions to errors) was selected for the present study purposes. This scale assesses perfectionism as a maladaptive coping strategy conceptualized within the Stress and Coping Cyclical Amplification Model of Perfectionism in Illness [46,47]. This subscale contains 9 items, and the internal consistency in our sample was of 0.87.

Pain intensity: As a relevant contextual variable, pain intensity in specific situations was used. In particular, we were interested in assessing said intensity in situations with high pain versus low pain. To do this, two items assessing maximum and minimum pain intensity during the last 7 days in a numerical rating scale were used (0 = no pain, and 10 = the worst pain you can imagine) [48,49]. The present scale provided scores for both the level of pain in high pain situations and the level of pain in low pain situations.

### 2.3. Data Analysis

First, descriptive and bivariate Pearson correlation analyses were performed. Secondly, we proceeded to divide the sample based on the disease severity criteria above mentioned. Next, the moderation analyses were conducted with model 1 from the PROCESS Macro version 3.4 [50]. Perfectionism was used as the moderator, pain as independent variable, and activity avoidance as the outcome. Both variables were centered before the analyses. Four models were tested, two for each of the disease severity groups (high vs. low/moderate), one for pain intensity in high pain situations, and one for pain intensity in low pain situations. Statistical significance was set at an alpha level of 0.05. In the PROCESS Macro, recommended values in conditional tables and graphical representations were the 16th, the 50th, and the 84th percentiles. Thus, these cut-offs were used to calculate conditional effects (i.e., effects of an independent variable on an outcome for different values of a moderator) when a moderation was found to be significant. In the post-hoc analyses, non-centered variables were used to facilitate the interpretation of results.

## 3. Results

### 3.1. Sociodemographic and Clinical Data of The Study

As can be seen in Table 1, the age of the participants ranged between 30 and 78 years, and the mean age was 56.91 years (SD = 8.9). Most of the participants were married or in a stable relationship (53%), had attended primary school (53.1%), and were mainly (32.5%) housewives or were retired (33.3%). Regarding clinical variables, the mean pain score was 7.15 (SD = 1.52; range 0−10), and the patients had experienced fibromyalgia for a mean of 12–14 years (SD = 8.45; range 1−46 years).

For the control of the sociodemographic variables considered, a previous analysis of the relationship of said variables with activity avoidance was carried out; no statistically significant relationship was found.

### 3.2. Means, Standard Deviations, and Pearson Correlations between Study Variables

Means, standard deviations, and Pearson correlations between study variables are presented in Table 2. Activity avoidance was positively associated with pain intensity in both high and low pain situations (*p* < 0.001) and with perfectionism (*p* < 0.001). No statistically significant associations were found between pain intensity (at high or low pain situations) and perfectionism. The effect sizes can be considered small (between 0.10–0.30), except in the case of the relationship between pain intensity in low pain situations and pain intensity in high pain situations, which presented a medium effect size (between 0.30–0.50).

### 3.3. Regression Including Moderation Analyses

The results of the regression analyses, including the analysis of moderation, for the two groups (high vs. low/moderate disease severity) in situations of high and low pain are presented in Table 3.

#### 3.3.1. High Severity Group

In the high severity group, the prediction of activity avoidance from pain intensity (in situations of high pain), perfectionism, and their interaction evidenced significant direct (positive) contributions of perfectionism (*p* < 0.001). The results also showed the moderation of perfectionism in the relationship between pain intensity and activity avoidance (Beta = −0.12, t = −3.19, *p* = 0.002, 95% CI −20, −0.05).

The prediction of activity avoidance from pain intensity (in low pain situations), perfectionism, and their interaction evidenced significant direct (positive) contributions of perfectionism (p < 0.001). The models predicting activity avoidance from pain intensity and perfectionism and their interaction explained 16% of the variance (in the case of pain in high pain situations) and 10% (in the case of pain in low pain situations).

#### 3.3.2. Low/Moderate Severity Group

In the low/moderate severity group, the prediction of activity avoidance from pain intensity (in high pain situations), perfectionism, and their interaction evidenced significant direct (positive) contributions of pain intensity (*p* = 0.040) and significant direct (positive) contributions of perfectionism (*p* = 0.005). The results also show a moderation of perfectionism in the relationship between pain intensity and activity avoidance (Beta = 0.06, t = 2.40, *p* = 0.018, 95% CI 0.01, 0.11).

The prediction of activity avoidance from pain intensity (in low pain situations), perfectionism, and their interaction evidenced significant direct (positive) contributions of perfectionism (*p* = 0.020). The models predicting activity avoidance from pain intensity and perfectionism and their interaction explained 14% of the variance (in the case of pain in high pain situations) and 7% (in the case of pain in low pain situations).

As noted earlier, post-hoc analyses were planned to analyze significant moderations more in-depth. These were computed and are presented in Table 4 and Figure 2 for the moderation of perfectionism in the relationship between pain intensity in high pain situations and activity avoidance in high severity group, and in Table 4 and Figure 3 for the moderation of perfectionism in the relationship between pain intensity in high pain situations and activity avoidance in low/moderate severity group.

As noted both in Table 4 and Figure 2 (high severity group), the contribution of pain intensity in high pain situations on activity avoidance varied at different values of perfectionism. Specifically, pain intensity in high pain situations was only significantly associated (positively) with activity avoidance when perfectionism was low (Beta = 1.238, *p* = 0.002, 95% CI = 0.454, 2.022). When perfectionism was high, no significant associations were found for pain intensity in high pain situations on avoidance of activity.

As noted both in Table 5 and Figure 3 (low/moderate severity group), the contribution of pain intensity in high pain situations on activity avoidance varied at different values of perfectionism. Specifically, pain intensity in high pain situations was only significantly associated (positively) with activity avoidance when perfectionism was high (Beta = 0.849, *p* = 0.002, 95% CI = 0.314, 1.383).

## 4. Discussion

The present study aimed to explore the relationship between perfectionism and activity avoidance within the framework of the psychological flexibility model, taking into account two contextual variables—one general contextual variable, disease severity (differentiating between high and low/moderate severity), and a specific contextual variable by taking into account high and low pain situations. According to the above-mentioned psychological flexibility model, our results show differential findings based on the performance of said specific contextual variable (high and low pain situations) in the more general contextual variable (disease severity).

On the one hand, when facing low pain situations, among both groups of high and low/moderate disease severity, the prediction of avoidance of activity is based on their perfectionism levels (the higher the perfectionism is, the higher the avoidance is), not having found any influence of pain intensity. Living with a chronic illness such as fibromyalgia creates an interference between attaining a pain controlling goal and other goals which are relevant to the person suffering chronic pain [51,52]. In low pain situations, this interference does not occur, therefore, the patient can pay attention to other relevant goals unrelated to pain. Attaining these relevant goals uninfluenced by pain could be determined by perfectionism, therefore creating a specific pattern in which these patients assess activities/goals and carry them out [53]. In this sense, the association between perfectionism and avoidance could be understood if we consider perfectionism to be a concern over mistakes [53,54].

On the other hand, when facing high pain situations, differential effects of pain intensity and of the pain intensity–perfectionism interaction on activity avoidance were observed as a function of the degree of disease severity. When pain is high, there is a conflict between attaining a goal aimed towards controlling pain and other goals that could be relevant to the patient [51,52]. Said conflict also exists in a wider context, which is dictated by the severity of the disease. Therefore, when this conflict occurs in a low disease severity context, a significant and positive association is found between pain intensity and avoidance, probably because (high) pain intensity is unusual in their day to day, therefore, they tend to avoid activity to avoid further pain, according to the fear of movement model [23]. In addition, the interaction effects show that, only among high perfectionist patients, a significant and positive relationship is observed between pain intensity and activity avoidance. It is possible that, in a broader context of low impact of disease, highly perfectionist patients visualize other occasions (e.g., less pain) to carry out the activity guaranteeing success according to their personal standards. Conversely, when this interference occurs in a high disease severity context, only significant effects of the pain intensity–perfectionism interaction are observed, and the direction is the opposite to that observed in low disease severity context. Specifically, in high disease severity context, only in non-perfectionist patients, a significant and positive relationship is observed between pain intensity and activity avoidance. This could be explained because patients with high disease severity may perceive that situational contexts (e.g., less painful situations) are less likely, which could be more favorable to carrying out the behaviors. Therefore, activity avoidance has a direct relationship with intensity of pain only among patients who are not perfectionists. Finally, our results show perfectionism has a direct and positive effect on activity avoidance among both high and low/moderate disease severity groups, regardless of whether they experienced low or high pain situations. In previous literature, perfectionism has been considered as a premorbid condition that implies that an individual is likely to focus on direct action and achievement. Women with fibromyalgia show extreme forms of perfectionism [55], which is also associated with maladaptive coping of the illness [56,57,58]. To the best of our knowledge, the specific association between perfectionism and activity avoidance has not been studied in chronic pain populations. Nevertheless, among healthy people, maladaptive perfectionism has been associated with avoidance of physical activity [59]. Another finding that should be highlighted is that the direct effect of perfectionism on avoidance is significant, independently of the patient disease severity, therefore, its negative effects could be generalized considering the heterogeneity of the patients [38,55]. In this sense, perfectionism is shown to be a variable with a direct positive effect on activity avoidance when controlling for pain intensity and disease severity.

These results support the need to accept the influence of psychological variables in chronic pain from the perspective of contextual models such as the psychological flexibility model [26]. Specifically, researchers have focused on exploring activity patterns in relation to situational factors (e.g., the existence of pain or an ongoing task) and personal goals. Psychological flexibility implies the capacity to persist or to change behavior in a way that (a) includes conscious and open contact with thoughts and feelings, (b) appreciates what the situation affords, and (c) is guided by one’s goals and values [54]. The flexibility model explains that two sets of influences interact in how behavior is coordinated. First, environmental influences that include direct experience (including our sensory experience of the world—what we see, hear, smell, taste, and feel directly)—and second, cognitive influences that include verbal, language-based, or cognitive processes, such as rules, instructions, appraisals, expectations, judgments, stories, and products of mental analysis [60,61].

In this context, our results show, consistent with past research, that pain severity is important in the relationship between psychological factors and outcomes [41] and additionally point out the need to take into account said influence from the perspective of contextual variables which are wider, such as disease severity [42]. In particular, from the models of psychological flexibility, activity avoidance could be determined not only by psychological factors (perfectionism) but also by a combination of these variables and other contextual ones (i.e., disease severity and high/low pain intensity situations). Therefore, taking into account the models mentioned above and in line with previous research [62], our results show that reducing the intensity of pain is not always sufficient to ensure a successful level of activity among chronic pain patients. Thus, the reduction of maladaptive perfectionism is also necessary due to its direct relationship with activity avoidance, regardless of the contextual variables considered. Additionally, as we have been suggesting, these associations should be interpreted in situations of high and low pain and in the context of high and low/moderate disease severity, which would reinforce the need, suggested in previous literature, to establish differential profiles for fibromyalgia patients according to their disease severity [42,63]. The creation of profiles would allow for more detailed analysis of adaptive and maladaptive variables, because activity and cognitive patterns are not functional or dysfunctional per se, and their effects on disability are underlying to personal goals and contextual factors [44].

These findings could serve to make interventions more efficient. In all cases, regardless of the degree of severity, the reduction of non-adaptive perfectionism is shown as an essential objective for the prevention of disability. Cognitive behavioral therapy for perfectionism has been shown to be an effective treatment for chronic pain [64], reducing perfectionism levels, physical impact, and fatigue [65]. Further, acceptance commitment therapy (ACT) techniques to increase psychological flexibility have been shown to be a valid treatment to improve perfectionism and pain-related outcomes simultaneously due to the role of acceptance in chronic pain and perfectionism [66]. ACT treatment, in accordance with the model of cognitive flexibility that guided the present study, aims to promote acceptance of one’s private experiences and participation in activities to accomplish one’s value-based goals [67].

Additionally, the present study findings may also have important clinical implications in the field of personalized treatments. For example, for patients with high severity of disease at moments of high pain, perfectionism is a strategy that compels them to cope with activity, and therefore interventions should be directed towards learning healthy activity patterns and trying to not engage in patterns which include excessive persistence due to its negative health effects [68,69]. Cognitive behavioral therapy and exercise training could be suggested as ways to maintain a balance between excessive persistence and activity avoidance [70]. Among patients with low/moderate disease severity, perfectionism when pain intensity is high constitutes a maladaptive strategy associated positively with avoidance. Hence, CBT and ACT therapy could be appropriate pathways to decrease perfectionism levels [35,71].

This study certainly has some limitations. First, the associations must be interpreted according to the cross-sectional nature of the design, which does not allow inferences of causality. Second, this study is based on self-report data; therefore, some of the significant associations found may be due to shared method variance. Third, the present study used pain severity as an indicator of disease severity. Although pain is the main diagnostic symptom in patients with fibromyalgia [5,6,11] and has been strongly related to associated symptoms and functional limitation, when being used as an indicator of disease severity [9,10] as in the present study, the results obtained should be interpreted based on this choice. Likewise, although the sociodemographic variables considered have not shown any significant relationship with activity avoidance, it would be of interest to analyze the role of these variables in the perfectionism–activity avoidance relationship within contextual functional models. This approach could be considered as a line of future research. Last, the results are based on Spanish fibromyalgia patients and females only, therefore, future research will be needed to explore whether the findings are generalizable to males and other pain populations.

## 5. Conclusions

The present study corroborates the maladaptive role of perfectionism in its association with activity avoidance regardless of the degree of severity of the disease and the pain experienced in the situation. Additionally, and as the main novelty, it provides results on the different roles of said variable in the relationship between pain intensity and activity avoidance in samples with different degrees of disease severity. This relationship is equally affected by the magnitude of the pain experienced. Specifically in situations of high pain, in patients with high disease severity, perfectionism leads to the performance of the task despite the pain. In those same situations, in patients with low disease severity, perfectionism is associated with activity avoidance, probably under the expectation of better future conditions for success. These findings could have important clinical implications for practices in the field of psychological and interdisciplinary treatments and personalization in fibromyalgia patients [55,72].

## Figures and Tables

**Figure 1 ijerph-17-08442-f001:**
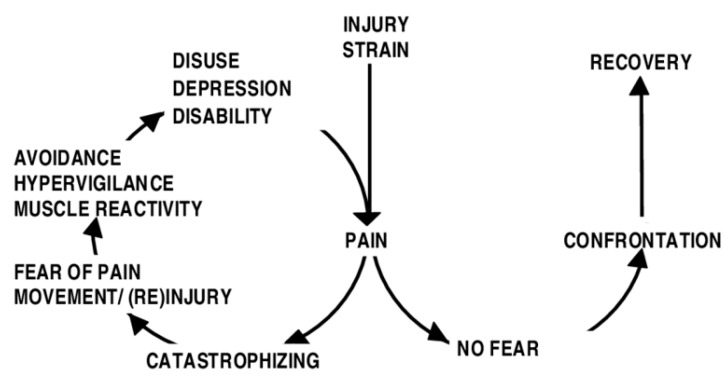
A cognitive-behavioral model of pain-related fear (based on Vlaeyen et al. [23]).

**Figure 2 ijerph-17-08442-f002:**
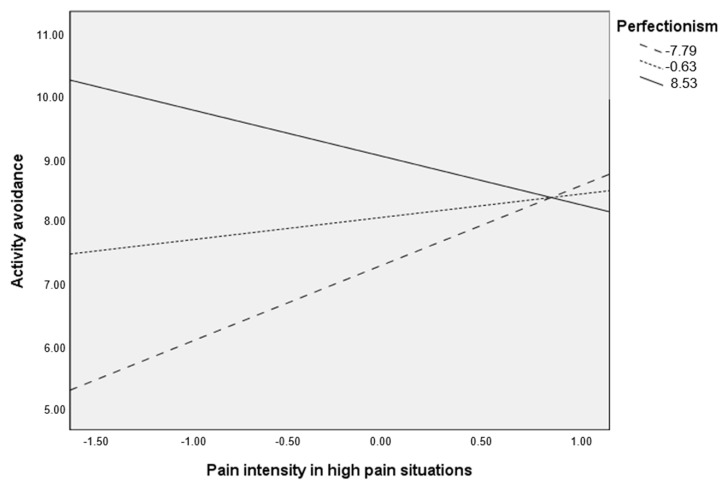
Relationship between pain intensity in high pain situations and activity avoidance at different levels of perfectionism in patients with high disease severity.

**Figure 3 ijerph-17-08442-f003:**
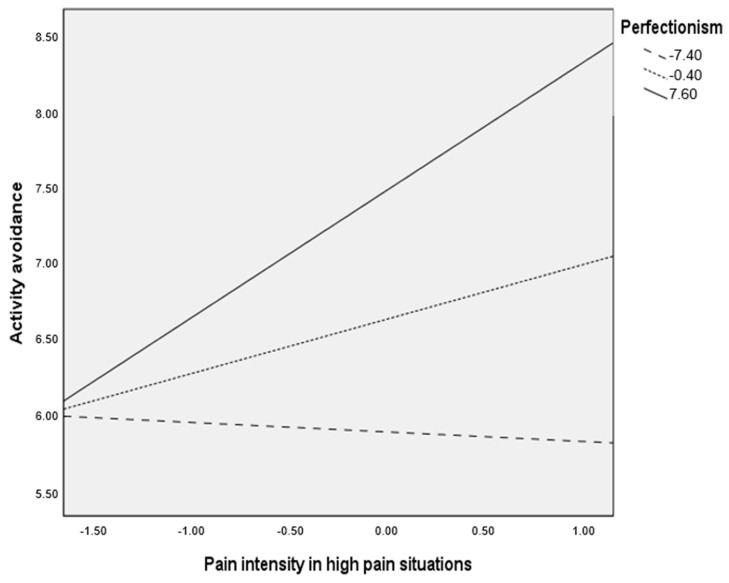
Relationship between pain intensity in high pain situations and activity avoidance at different levels of perfectionism in patients with low/moderate disease severity.

**Table 1 ijerph-17-08442-t001:** Sociodemographic data of the sample.

Variable	Fibromyalgia Women (n = 228)
Age, mean (SD)	56.91 (8.94)
Education level, n (%) No studies Primary Secondary University	31 (13.5) 121 (53.1) 61 (26.8) 15 (6.6)
Marital Status, n (%) Married Single Widowed/separated	121 (53) 25 (11) 82 (36)
Employment status, n (%) Currently employed Unemployed Sick leave Retired Retired due to disability Housewife	27 (11.8) 28 (11.9) 24 (10.5) 33 (14.5) 42 (18.8) 74 (32.5)
Pain, mean (SD)	7.15 (1.52)
Time from fibromyalgia diagnosis, mean (SD)	12.14 (8.45)

**Table 2 ijerph-17-08442-t002:** Means, standard deviations, and Pearson correlations between study variables.

Variable	Mean (SD)	2	3	4
1. Activity avoidance	7.45 (2.71)	0.27 **	0.24 **	0.29 **
2. Perfectionism	22.09 (7.65)		0.03	<0.01
3. Pain intensity in LPS	5.58 (2.09)			0.48 **
4. Pain intensity in HPS	8.44 (1.52)			

** *p* < 0.01; LPS, low pain situations; HPS, high pain situations.

**Table 3 ijerph-17-08442-t003:** Prospective prediction of activity avoidance from pain intensity (in high and low pain situations), perfectionism (concern over mistakes), and their interaction in high and low/moderate disease severity groups.

Variable	R^2^	F	*p*	Beta	t	*p*	95% CI
High disease severity group							
DV = Activity avoidance	0.16	7.71	<0.001				
Pain intensity in HPS				0.29	1.15	0.24	−20, 0.78
Perfectionism				0.10	4.05	<0.001	0.05, 0.15
Interaction				−0.12	−3.19	0.002	−0.20, −0.05
DV = Activity avoidance	0.10	4.37	0.005				
Pain intensity in LPS				0.14	1.25	0.21	−0.08, 0.37
Perfectionism				0.08	3.44	<0.001	0.04, 0.14
Interaction				<0.001	0.11	0.91	−0.02, 0.03
Low/moderate disease severity group							
DV = Activity avoidance	0.14	5.49	0.001				
Pain intensity in HPS				0.38	2.07	0.04	0.02, 0.76
Perfectionism				0.11	2.85	0.005	0.03, 0.18
Interaction				0.06	2.40	0.018	0.01, 0.11
DV = Activity avoidance	0.07	2.57	0.058				
Pain intensity in LPS				0.12	0.67	0.50	−0.24, 0.50
Perfectionism				0.09	2.36	0.02	0.01, 0.16
Interaction				0.03	1.14	0.25	−0.02, 0.08

High disease severity group (< 7) (*n* = 124); low/moderate disease severity group (> 7) (*n* = 104); LPS, low pain situations; HPS, high pain situations; DV, Dependent Variable.

**Table 4 ijerph-17-08442-t004:** Conditional effects of pain intensity in high pain situations on activity avoidance at values of perfectionism (high disease severity group).

Perfectionism	Beta (Pain Intensity in HPS)	t	*p*	95% CI
−7.78	1.238	3.127	0.002	0.454, 2.022
−0.63	0.363	1.451	0.149	−0.132, 0.859
8.53	−0.756	−1.880	0.062	−1.551, 0.04

**Table 5 ijerph-17-08442-t005:** Conditional effects of pain intensity in high pain situations on activity avoidance at values of perfectionism (low/moderate disease severity group).

Perfectionism	Beta (Pain Intensity in HPS)	t	*p*	95% CI
−7.40	−0.063	−0.240	0.810	−0.584, 0.458
−0.40	0.362	1.942	0.054	−0.008, 0.732
7.59	0.849	3.148	0.002	0.314, 1.383

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
