# Peer review of "Perfectionism and Pain Intensity in Women with Fibromyalgia: Its Influence on Activity Avoidance from The Contextual Perspective"

_ijerph, 2020, doi:10.3390/ijerph17228442_

Round 1

Reviewer 1 Report

The work is very good and I recommend your publication. Some suggestions are:

  • In the introduction it would be appropriate to indicate examples of studies on associated pathologies (cognitive decline, sleep disturbances, etc.). For example: https://pubmed.ncbi.nlm.nih.gov/25456469/ 
  • In the objectives of the introduction, specify that it is about patients with fibromyalgia.

  • In the introduction, differentiate the hypotheses and objectives of the study more clearly (such as writing this information in a different paragraph).

  • Include more recent bibliography. For example:

    https://ard.bmj.com/content/75/Suppl_2/1190.3

    https://www.sciencedirect.com/science/article/abs/pii/S0191886918304458

    https://pubmed.ncbi.nlm.nih.gov/30858740/

Author Response

Response to Reviewer 1 Comments

First of all, we appreciate the time that you and the other reviewers have dedicated to reading the manuscript and providing suggestions. We believe your suggestions have enriched the manuscript considerably. We have incorporated all the comments and we hope the new changes meet your expectations, and answer the points you have made.

Point 1: In the introduction it would be appropriate to indicate examples of studies on associated pathologies (cognitive decline, sleep disturbances, etc.). For example: https://pubmed.ncbi.nlm.nih.gov/25456469/ 

Response 1: Following the reviewer’s comment, we have added three references, according to each pathology (page 1, lines 38-41) and erased a previous one.

Point 2: In the objectives of the introduction, specify that it is about patients with fibromyalgia.

Response 2: Following the reviewer’s comment we have added in Page 4, lines 121 and 123 that the study is with patients with fibromyalgia.

Point 3: In the introduction, differentiate the hypotheses and objectives of the study more clearly (such as writing this information in a different paragraph).

Response 3: This information has been differentiated in separate paragraphs. In addition, hypotheses have been added (Page 4, lines 123-130).

Point 4: Include more recent bibliography. For example:  

https://ard.bmj.com/content/75/Suppl_2/1190.3

https://www.sciencedirect.com/science/article/abs/pii/S0191886918304458

https://pubmed.ncbi.nlm.nih.gov/30858740/

Response 4: Following the reviewer’s comment and the examples suggested, we have added two references:

  • Pinto, A. M., Pereira, A. T., Costa, C., Marques, M., da Silva, J. A. P., & Macedo, A. (2016). Perfectionism in Chronic Pain: Are There Differences between fibromyalgia, Rheumatoid Arthritis and Healthy Controls? (page 3. Line 84)
  • Sirois, F. M., Toussaint, L., Hirsch, J. K., Kohls, N., Weber, A., & Offenbächer, M. (2019). Trying to be perfect in an imperfect world: A person-centred test of perfectionism and health in fibromyalgia patients versus healthy controls. Personality and Individual Differences137, 27-32. (page 3. Line 84).

The third example suggested had been cited already (Galvez-Sánchez, C. M.; Duschek, S.; Reyes del Paso, G. A. Psychological impact of fibromyalgia: current perspectives. Psychol. Res. Behav. Manag. 2019, 12, 117–127. DOI: 10.2147/PRBM.S178240). (Page 1, line 39).

Reviewer 2 Report

Perfectionism and pain intensity in women with fibromyalgia: its influence on activity avoidance from the contextual perspective.

This paper describes a cross-sectional study of women with fibromyalgia and the associations between perfectionism, activity avoidance, and two pain-related contextual factors (pain intensity and disease severity). As the authors report, few studies have examined perfectionism as a psychological variable that may influence the relation between pain and avoidance in chronic pain populations. The exposition of this research would benefit from the authors’ consideration of the following points.

Introduction

  • In general, the manuscript would benefit from a clear outline into psychological flexibility and role of perfectionism.
  • There were times in which the introduction was somewhat hard to read due to run-on sentences.
  • Last sentence of first paragraph of Introduction (lines 41-43) makes mention of “pain (disease severity).” The idea that disease severity in fibromyalgia is classified based on self-reported pain intensity/severity is not discussed until the Method section. It would be helpful to include more detail as to the use of this as a measure of disease severity given that pain intensity is very subjective and not always a measure of how “severe” a pain diagnosis is for a person.
  • Explanation of fear of movement/reinjury framework (lines 50-52) may not be necessary, as the name of the framework is self-explanatory.
  • The explanation of the self-perpetuating cycle of fear, avoidance, and functional impairment might be enhanced by a visual aide.
  • Sentence in lines 55-57 needs grammatical revision.
  • It would be helpful to include a more robust literature review or theoretical explanation of perfectionism as a variable that might connect to pain processes/outcomes.
    • For example, what are the specific hypotheses regarding perfectionism as a moderator between pain intensity and activity avoidance?
    • There is literature on the role of perfectionism in youth with chronic pain (ET Randell) that could aid the authors in framing up the introduction.
  • Utilizing subheadings may also help organize the introductory argument.

Method

  • Clarify if participants were compensated for participation.
  • Please mention early in the methods that the study was conducted in a Spanish speaking country.
  • Again there is a concern that pain is representing disease severity especially given that pain is subjective. More evidence on this decision is needed.

Results

  • Use of “high pain” (e.g., line 193) is confusing; it would be more clear the term “pain (in high pain situations)” was used consistently throughout Results section.
  • The author may wish to include a power analysis, especially if interpreting a finding that is not statistically significant (line 196).
  • Could age be considered a covariate? Is it possible findings would differ depending on patient age?
  • Visual aids would be very helpful in representing the moderation models that were tested.
  • Effect size estimates are needed.

Discussion

  • Overall, the discussion is very hard to follow. There appears to be times in which the authors say opposite messages. For example on page 9 line 254 there is mention of pain intensity not influencing patients, but then on the same page line 291 it was noted that pain severity is important.
  • Lines 227 and 238 – does it make sense to interpret this finding in light of the moderation analyses (i.e., can we say that perfectionism has an overall direct, positive effect on activity avoidance if this was not the case for the high severity group when they experienced low pain)? Could perhaps consider tempering discussion of this finding.
  • Line 237 – “fact” might be too strong a word; perhaps “finding” would work better.
  • Line 321 – typo, “excessive persistence” typed twice in a row.
  • Final “Conclusion” paragraph makes an interesting distinction, discussing how perfectionism can be maladaptive for both high and moderate/low severity disease groups, but for different reasons. It would be interesting to see this fleshed out more in the discussion. Also only mentioned CBT and ACT in the conclusions section, but not in the actual discussion.

Author Response

Response to Reviewer 2 Comments

First of all, we appreciate the time that you and the other reviewers have dedicated to reading the manuscript and providing suggestions. We believe your suggestions have enriched the manuscript considerably. We have incorporated all the comments and we hope the new changes meet your expectations and answer the points you have made.

INTRODUCTION

Point 1: In general, the manuscript would benefit from a clear outline into psychological flexibility and role of perfectionism

Response 1: Following the reviewer’s comment has been added, we have now included a description of the role perfectionism plays within the model of psychological flexibility.

Page 3, Lines 85-93:

Based on this model, perfectionism is defined by cognitive rigidity and behavioral inflexibility [32 – 33], where the person presents lack of situation-strategy fit and uses the same strategies [34], regardless of the context. One of the underlying mechanisms in perfectionism is experiential avoidance [35], which influences health outcomes. For this reason, acceptance and commitment therapy has been used [33, 35] to reduce perfectionism and the associated symptoms. Previous studies have shown that there is an association between perfectionism and stress, poor mental health, reductions in functioning and the frequency/intensity of pain and fatigue [36 -38]. Particularly, perfectionism is associated with reductions in health functioning in women with fibromyalgia [37], mediated by behavioral disengagement processes, denial, and self-blame [29 - 30].

Point 2: There were times in which the introduction was somewhat hard to read due to run-on sentences

Response 2: We are aware of the importance of the writing aspects of the article and we thank the reviewer very much for this comment. Thus, we have revised the manuscript again paying attention to this issue, correcting the run-on sentences and simplifying the writing of the manuscript. Changes have been made to the lines:

  • Page 2, Lines 51-53: The complex symptomatology of fibromyalgia, including pain (disease severity), interferes in patients’ daily function, preventing them from accomplishing their goals and reducing their ability to plan work and social activities [13].
  • Page 2, Lines 55-59: Fibromyalgia treatments that have included activities and physical exercise as one of their therapeutic aims have shown positive effects on patients’ health outcomes [16 - 18]. However, people with fibromyalgia often show lower levels of physical activity and greater sedentary behaviors [19 - 21] due to pain and fatigue that are often initially exacerbated with increased activity [22].
  • Page 3, Lines 75-78: This model postulates that there is no linearity between two variables and even that the modulators themselves can depend on contextual variables, that influence committed actions towards goals. Therefore, knowing the context in which the behavior occurs is important to determine its influence on outcomes functioning [26].
  • Page 12, Lines 360-363: To the best of our knowledge, the specific association between perfectionism and activity avoidance has not been studied in chronic pain populations. Nevertheless, among healthy people, maladaptive perfectionism has been associated with avoidance of physical activity [59].
  • Page 14, Lines 464-467: Among patients with low/moderate disease severity, perfectionism, when pain intensity is high, constitutes a maladaptive strategy, associated positively with avoidance. Hence, CBT and ACT therapy could be an appropriate alternative to decrease perfectionism levels [71 - 72].

Point 3: Last sentence of first paragraph of Introduction (lines 41-43) makes mention of “pain (disease severity).” The idea that disease severity in fibromyalgia is classified based on self-reported pain intensity/severity is not discussed until the Method section. It would be helpful to include more detail as to the use of this as a measure of disease severity given that pain intensity is very subjective and not always a measure of how “severe” a pain diagnosis is for a person

Response 3: We thank the reviewer for this comment. Pain as the main symptom in fibromyalgia is revealed in the initial diagnostic criteria [5] that have evolved, and currently other additional symptoms are also contemplated (i.e. fatigue, non-restorative sleep, cognitive disorders) in the recent diagnostic criteria [6, 11]. Both diagnostic criteria continue to coexist today and previous literature has shown a good agreement between them for FM diagnosis [5 – 6, 12]. Although pain has been strongly related to associated symptoms and functional limitation, being used as an indicator of disease severity [9 – 10] as in the present study, it is still one possible indicator, of the numerous existing ones, of the severity of the disease (Page 1 and 2, lines 42-44).

For this reason, as suggested by the reviewer, the use of pain severity as an indicator of disease severity has been justified in the introduction of the manuscript (Page 1 and 2, lines 42-44). Additionally, after this theoretical justification, we have proceeded to review, throughout the manuscript, that it is specified that pain has been considered as an indicator of the severity of the disease. Finally, in the section on limitations, we have also proceeded to mention this issue and the limitations it implies for the generalization of the results (Page 14, lines 471-478).

Point 4: Explanation of fear of movement/reinjury framework (lines 50-52) may not be necessary, as the name of the framework is self-explanatory.

Response 4:  Indeed, as indicated by the reviewer, the information provided in these lines is redundant. It has been eliminated.

Point 5: The explanation of the self-perpetuating cycle of fear, avoidance, and functional impairment might be enhanced by a visual aide.

Response 5: Following the reviewer’s recommendation, we added Figure 1. A cognitive-behavioral model of pain-related fear (based on Vlaeyen et al. [23] (Page 2, Lines 70-71) in the Introduction.

Point 6: Sentence in lines 55-57 needs grammatical revision.

Response 6: We totally agree, the phrase has been reformulated as follows in Page 2, lines 66-69: Therefore, it could be said that a self-perpetuating cycle is created, based on prolonged avoidance of motor activities and pain avoidance goals. This behavioral pattern is a maladaptive way of coping with pain [24], this can have harmful consequences, both physically and psychologically [25] (Page 2, lines 66-69).

Point 7: It would be helpful to include a more robust literature review or theoretical explanation of perfectionism as a variable that might connect to pain processes/outcomes.

Response 7: Following the reviewer’s recommendation, we have carried out a new literature search and have found two studies on perfectionism in pain and health outcomes [39 – 40]. The scarcity of this type of study has also been pointed out in the manuscript (Page 3, lines 94-97).

Point 8: For example, what are the specific hypotheses regarding perfectionism as a moderator between pain intensity and activity avoidance?

Response 8: Thank you for your comment. The corresponding hypotheses have been specified.

Point 9: There is literature on the role of perfectionism in youth with chronic pain (ET Randell) that could aid the authors in framing up the introduction

Response 9: Thank you for your guide. The two studies found in this regard by this author have been very useful (Page 3, lines 94-97).

Point 10: Utilizing subheadings may also help organize the introductory argument.

Response 10: Following the suggestion of the reviewer, two subheadings have been incorporated to help organize the introductory argument:

Activity avoidance, fear of movement and disability (Page 2, line 50)

Perfectionism and fibromyalgia. A contextual-functional approach (Page 2, line 72)

METHOD

Point 1: Clarify if participants were compensated for participation.

Response 1: This information has been added in the second paragraph of the Materials and Method section (Page 4, lines 140-141).

Point 2: Please mention early in the methods that the study was conducted in a Spanish speaking country.

Response 2: It has been noted that the study was conducted in a Spanish speaking country (Page 4, line 135).

Point 3: Again there is a concern that pain is representing disease severity especially given that pain is subjective. More evidence on this decision is needed

Response 3: We agree with the reviewer on the need to argue the choice of pain severity as an indicator of disease severity. Following the argument set out in the introduction, we have proceeded to justify the choice of this indicator over other indicators that could also be used. In any case, it is also pointed out that the results must be interpreted taking this decision into account (Page 4, lines 155-156).

RESULTS

Point 1: Use of “high pain” (e.g., line 193) is confusing; it would be more clear the term “pain (in high pain situations)” was used consistently throughout Results section.

Response 1: We agree with the reviewer in this comment. We have replaced the term "high / low pain" with the term "pain intensity in high / low pain situations" throughout the manuscript.

Point 2: The author may wish to include a power analysis, especially if interpreting a finding that is not statistically significant (line 196).

Response 2: We thank the reviewer for this comment, which has made us reflect on whether or not it is really appropriate to present a non-significant result as a tendency. Finally, we have come to the conclusion that it is not the best option, which has made us correct the sentence and present that relationship as non-significant. However, the comment showed us the convenience of including a Power Analysis that allows us to verify that the sample used in each group (high vs moderate / low severity) is sufficient to carry out the analyzes performed. We have chosen to add this information in the section Participants (Page 4, lines 163-167).

Point 3: Could age be considered a covariate? Is it possible findings would differ depending on patient age?

Response 3:   Thanks for the interesting suggestion. Age was not considered as a covariate, because the preliminary analyses did not show a significant relationship between activity avoidance and any of the sociodemographic variables considered, including age. This additional information has been incorporated in the Results section (Page 6, lines 219-223). However, the possible influence of sociodemographic and clinical variables in the relationship between perfectionism and activity avoidance constitutes an object of study in itself, so they could be addressed in future studies. This suggestion has been incorporated as a line of future research (Page 14, lines 475-478).

Point 4: Visual aids would be very helpful in representing the moderation models that were tested.-.

Response 4: We consider that Figure 2 and Figure 3 may be useful as a visual aid of the effects of perfectionism on the pain intensity-activity avoidance relationship. We are open to any other type of suggestion regarding graphs/figures that the reviewer wants to make (Page 10, lines: 295-297; Page 11, lines 310-312).

Point 5: Effect size estimates are needed.

Response 5: Thanks for the comment. We have proceeded to comment effect sizes of both the correlation analyzes (Page 7, lines 233-236) and the moderation analysis (Page 4, lines 166).

DISCUSSION

Point 1: Overall, the discussion is very hard to follow. There appears to be times in which the authors say opposite messages. For example on page 9 line 254 there is mention of pain intensity not influencing patients, but then on the same page line 291 it was noted that pain severity is important.

Response 1: Thanks for your suggestion. Indeed, the discussion, as it was organized, was difficult to follow. The lack of clarity led, as the reviewer points out, to seemingly contradictory messages. That is why the discussion has been completely restructured. In particular, the results of moderation and its differential results based on the degree of disease severity have been discussed first. Second, the direct effects of perfectionism have been discussed. Finally, the limitations are exposed and the results are summarized based on the models of psychological flexibility, pointing out the practical implications. (Pages 11-15.

Point 2: Lines 227 and 238 – does it make sense to interpret this finding in light of the moderation analyses (i.e., can we say that perfectionism has an overall direct, positive effect on activity avoidance if this was not the case for the high severity group when they experienced low pain)? Could perhaps consider tempering discussion of this finding.

Response 2: Thanks again for the suggestion. When carrying out a complete restructuring of the discussion, we consider that the interpretation problems associated with these lines have been corrected.

Point 3: Line 237 – “fact” might be too strong a word; perhaps “finding” would work better (line 321).

Response 3: Thanks for the comment. The change has been made. 

Point 4: Line 321 – typo, “excessive persistence” typed twice in a row.

Response 4: Thanks. The repetition has been eliminated.

Point 5: Final “Conclusion” paragraph makes an interesting distinction, discussing how perfectionism can be maladaptive for both high and moderate/low severity disease groups, but for different reasons. It would be interesting to see this fleshed out more in the discussion. Also only mentioned CBT and ACT in the conclusions section, but not in the actual discussion.

Response 5: Following the reviewer's suggestion, the practical implications of the results have been discussed in greater detail in the discussion section (Page 14, Lines 458-467).

Reviewer 3 Report

Thank you for giving me the possibility to review this manuscript, dealing with a very interesting topic. However, before considering it for publication on IJERPH, the following points should be addressed:

  1. please perform an extensive English revision (improve the syntax!)
  2. Abstract: should be improved. Please add more data
  3. M&M: how perfectionism has been defined???
  4. Results: please add a table summarizing the main data of the study (mean age; gender; time from fibromyalgia diagnosis; comorbidities; drugs assumption; smocking status...)
  5. discussion: should be improved. Please comment about the relationship between the sample features (mean age; gender; time from fibromyalgia diagnosis; comorbidities; drugs assumption; smocking status...) and questionnaire results.

Author Response

Response to Reviewer 3 Comments

First of all, we appreciate the time that you and the other reviewers have dedicated to reading the manuscript and providing suggestions. We believe your suggestions have enriched the manuscript considerably. We have incorporated all the comments and we hope the new changes meet your expectations and answer the points you have made.

Point 1: please perform an extensive English revision (improve the syntax!).

Response 1: Thank you for the suggestion. The text has been revised by a bilingual expert in the field who has extensive experience with editing and proofreading manuscripts in this field.

Point 2: Abstract: should be improved. Please add more data.

Response 2: Thank you for your recommendation. Significance data have been incorporated for each of the analyses performed (Page , lines 25-29). We consider that the information is now complete; incorporating additional information would mean exceeding the limits of the journal in relation to the number of words in the abstract.

Point 3: M&M: how perfectionism has been defined???

Response 3: Thank you for your comment. The definition of perfectionism used has been added to Page 5, lines 188-190. In addition, it has been specified that the sub-dimension used refers specifically to: “Perfectionism is the belief that perfection can and should be achieved. Perfectionists value themselves based on their self-imposed achievements. Therefore, the Concern over mistakes subscale (reflecting negative reactions to errors) was selected for the present study purposes.”

Point 4: Results: please add a table summarizing the main data of the study (mean age; gender; time from fibromyalgia diagnosis; comorbidities; drugs assumption; smocking status...).

Response 4: Thanks for your suggestion, we have proceeded to add an explanatory table of the sociodemographic and clinical data of the study (Table 1, page 6-7, lines 224-225. Given that these data were explained in the section on participants in the method, the results section has been changed, dedicating a heading for it "3.1. Sociodemographic and clinical data of the study" Page 6, lines 218-223.

Point 5: discussion: should be improved. Please comment about the relationship between the sample features (mean age; gender; time from fibromyalgia diagnosis; comorbidities; drugs assumption; smocking status...) and questionnaire results.

Response 5: Thank you for your comment. Indeed, a prior analysis was necessary to know and control the possible effect of socio-demographic variables on activity avoidance. These analyses did not show a significant relationship between activity avoidance and any of the sociodemographic variables considered, so this topic has not been considered for discussion. However, this additional information has been incorporated in the Results section (Page 7, lines 225-227). Nevertheless, the possible influence of sociodemographic and clinical variables in the relationship between perfectionism and activity avoidance constitutes an object of study in itself, so they could be addressed in future studies. This suggestion has been incorporated as a line of future research (Page 14, lines 475-478).

Round 2

Reviewer 2 Report

Overall, the authors addressed the concerns raised by all the reviewers and the resulting manuscript is much improved.

Reviewer 3 Report

 The article is suitable of publication on IJERPH in the present form. All the suggested changes have been made in the revised manuscript.